# Neural Stem Cells Overexpressing Arginine Decarboxylase Improve Functional Recovery from Spinal Cord Injury in a Mouse Model

**DOI:** 10.3390/ijms232415784

**Published:** 2022-12-13

**Authors:** Yu Mi Park, Jae Hwan Kim, Jong Eun Lee

**Affiliations:** 1Department of Anatomy, Yonsei University College of Medicine, Seoul 03722, Republic of Korea; 2BK 21 PLUS Project for Medical Science, Yonsei University College of Medicine, Seoul 03722, Republic of Korea; 3CHA Advanced Research Institute, CHA University, CHA Bio-Complex, 335, Pangyo-ro, Bundang-gu, Seongnam-si 13488, Gyeonggi-do, Republic of Korea; 4Department of Biomedical Science, CHA University, CHA Bio-Complex, 335, Pangyo-ro, Bundang-gu, Seongnam-si 13488, Gyeonggi-do, Republic of Korea; 5Brain Research Institute, Yonsei University College of Medicine, Seoul 03722, Republic of Korea

**Keywords:** spinal cord injury, functional recovery, neurogenesis, glial scar, axonal re-myelination, neural progenitor cells, cell transplantation, arginine decarboxylase, agmatine, gene therapy

## Abstract

Current therapeutic strategies for spinal cord injury (SCI) cannot fully facilitate neural regeneration or improve function. Arginine decarboxylase (ADC) synthesizes agmatine, an endogenous primary amine with neuroprotective effects. Transfection of human ADC (hADC) gene exerts protective effects after injury in murine brain-derived neural precursor cells (mNPCs). Following from these findings, we investigated the effects of hADC-mNPC transplantation in SCI model mice. Mice with experimentally damaged spinal cords were divided into three groups, separately transplanted with fluorescently labeled (1) control mNPCs, (2) retroviral vector (pLXSN)-infected mNPCs (pLXSN-mNPCs), and (3) hADC-mNPCs. Behavioral comparisons between groups were conducted weekly up to 6 weeks after SCI, and urine volume was measured up to 2 weeks after SCI. A subset of animals was euthanized each week after cell transplantation for molecular and histological analyses. The transplantation groups experienced significantly improved behavioral function, with the best recovery occurring in hADC-mNPC mice. Transplanting hADC-mNPCs improved neurological outcomes, induced oligodendrocyte differentiation and remyelination, increased neural lineage differentiation, and decreased glial scar formation. Moreover, locomotor and bladder function were both rehabilitated. These beneficial effects are likely related to differential BMP-2/4/7 expression in neuronal cells, providing an empirical basis for gene therapy as a curative SCI treatment option.

## 1. Introduction

Spinal cord injury (SCI) currently has no curative therapy. Patients with SCI immediately lose all subdamaged motor, sensory, and autonomic nervous system functions. Secondary processes occurring at the injury site can worsen SCI [1,2]. Furthermore, bladder sphincter relaxation is absent, leading to urinary retention [3].

Characterized by an initial mechanical trauma to the spinal cord, SCI results in breakdown of the blood–brain barrier, activation of glial cells, and necrosis [4]. A secondary signaling cascade causes a cyclic increase in inflammatory cytokines that leads to apoptosis and progressive oligodendrocyte loss, eventually resulting in demyelination and axonal degeneration [5]. Moreover, the adult central nervous system (CNS) has poor trophic support and a growth inhibitory environment that is hostile to endogenous spinal cord regeneration [6,7]. 

After experiencing SCI, novel circuits are formed from both damaged and undamaged axons through axonal branching, dendritic growth, and reorganization [8,9]. Thus, SCI results in a highly reactive environment that presents significant obstacles for repair, lowers drug efficacy, and hampers the survival and integration of transplanted cells [1,8]. The main goal of treatment after SCI is to recover damaged motor and sensory axons, reverse demyelination, and heal glial scars [9,10]. Cell transplantation remains the most promising strategy for achieving these goals and can potentially restore function by replacing cells and restoring neuronal circuitry [10,11,12,13,14,15]. Grafted cells can provide trophic support for neurons and manipulate the environment within the damaged spinal cord to facilitate axonal regeneration or promote plasticity [9,11,12]. Sources for cell transplantation are typically multipotent stem cells, including embryonic stem cells (ESCs) [15,16], neural stem cells (NSCs), neural precursor cells (NPCs) [13,17,18,19], induced pluripotent stem cells (iPSCs) [20], and mesenchymal stem cells (MSCs) [21]. Of these, NSCs and NPCs have been the most effective in treating CNS diseases [13,17,22]. Regardless of cell type, stem cell-mediated therapy faces the same major challenge of controlling the survival, integration, and differentiation of transplanted cells [14]. 

Endogenous NPCs are multipotent cells with the potential to differentiate into neuronal and glial lineages within the CNS [13,17]. Experiments in demyelinated animal models have shown that intraspinally or intraperitoneally transplanted NPCs can integrate into damaged tissue, differentiate into myelinating oligodendrocytes, and cause clinical improvement [12,13,17]. Similarly, NPC transplantation into the SCI site results in integration and functional recovery, with the transplanted cells differentiating into neurons, astrocytes, and oligodendrocytes [12,13,23,24]. 

The introduction of gene therapy, growth factors, and anti-inhibitors has provided new ways to modify maladaptive transcription in a cell or to introduce novel genes [1,25]. Simultaneously delivering therapeutic genes with stem cells post-SCI may increase the viability of damaged cells and promote their regeneration, while also limiting inflammation, demyelination, and astrogliosis [26,27]. Therefore, successful SCI treatment must consider various forms of stem cell transplantation.

One compound that could enhance the effectiveness of stem cell transplants is the cation agmatine, formed from L-arginine decarboxylation by the mitochondrial enzyme arginine decarboxylase (ADC) [28]. Application of exogenous agmatine exerts neuroprotective effects in multiple injury models [29,30,31,32]. Several characteristics of agmatine suggest that it is involved in mammalian neurotransmission. First, agmatine is locally synthesized by neurons in the brain [33]. Second, it is stored in heterogeneously distributed neurons and astrocytes [34,35,36]. Third, agmatine is packaged into synaptic vesicles of neurons and released from axon terminals upon neuronal depolarization [37,38]. Other evidence of agmatine’s neuromodulator role is the presence of a specific cellular-uptake mechanism and metabolic enzyme (agmatinase) [30].

Agmatine treatment minimizes postischemic neuroinflammation by suppressing injury-induced deleterious alterations in the immune system [30,39,40]. Recombinant hex histidine ADC (hisADC) induces endogenous agmatine synthesis during stress in vitro [41,42], and ADC overexpression attenuates the oxidative burden in mouse cortical neural stem cells [43]. In addition, agmatine treatment of NPCs from the subventricular zone increased neurogenesis and suppressed astrogenesis, while ADC gene transfection increased NPC neuronal differentiation [42]. A relationship between agmatine and ADC was further demonstrated with findings that ADC messages are expressed in brain regions, ADC is regulated in neuronal cells, and siRNA downregulation of ADC activity lowers agmatine production [44]. Lastly, when human mesenchymal stromal cells (hMSCs) were overexpressed with hisADC after SCI and transplanted into the patient, behavioral improvement was observed [45].

Agmatine plays an important role in the production and inhibition of neuronal and glial cells after SCI through its regulation of bone morphogenetic protein (BMP)-2/4/7 expression. Specifically, agmatine administration post-SCI increases BMP-2/7 expression in neurons and oligodendrocytes while decreasing BMP-4 expression in astrocytes [43,46]. Bone morphogenetic proteins (BMPs) refer to approximately 15 growth-regulating polyfunctional cytokines from the transforming growth factor beta (TGF-β) superfamily; they are widely expressed in both intact and injured spinal cords [46,47,48]. Previous studies using SCI animal models have suggested that BMP-2/4/7 have similar expression patterns in neurons and neuroglial cells. Moreover, expression patterns are closely related to improving motor function [46,49]. Normally expressed in the intact spinal cord, various BMP ligands and receptors are rapidly up- or downregulated after injury. The functions of several BMPs are well studied. For instance, BMP-4 promotes astrocytic differentiation while inhibiting neuronal and oligodendrocyte differentiation [50,51]. BMP-7 inhibits oligodendrocyte cell death and increases neuronal survival in SCI [52,53,54]. 

In this context, we hypothesized that transplanting recombinant human ADC-murine-cortex-derived NPCs (hADC-mNPCs) is a more effective SCI treatment than current methods. We therefore sought to explore NPC-based combination approaches that emphasize ADC gene therapy for reconstructing damaged spinal neural circuits and improving SCI recovery. Additionally, because agmatine and ADC are strongly linked, we aimed to confirm whether ADC is also involved in controlling BMP expression after SCI. Our study should greatly benefit the development of curative therapy for these debilitating injuries.

## 2. Results

### 2.1. Treatment with hADC-mNPC Led to Enhanced Cell Survival around the SCI Lesion Site

The results of PKH-26 staining indicated that mNPCs, pLXSN-mNPCs, and hADC-mNPCs were clustered together and migrated from the injection site (thoracic vertebra [T]8 and T10) to the lesion site (T9) at around 1 week post-transplantation (Figure 1A). The PKH-26 signal was significantly higher at the epicenter in hADC-mNPC-transplanted mice than in mNPC-transplanted mice or pLXSN-mNPC-transplanted mice (Figure 1B). Additionally, the hADC-mNPC group had significantly higher fluorescence intensity than the mNPC and pLXSN-mNPC groups (Figure 1B).

### 2.2. Transplantation of hADC-mNPCs Attenuated Astrocyte Clustering at the Lesion Site

We used GFAP immunoreactivity to quantify astrocytes in mNPC, pLXSN-mNPC, and hADC-mNPC mouse tissue sections. By 2 weeks post-transplantation, hADC-mNPC mice had a significantly smaller GFAP-positive area at the lesion site than mNPC and pLXSN-mNPC mice (Figure 2A). The hADC-mNPC group also had a significantly smaller glial scar area than the mNPC and pLXSN-mNPC groups (Figure 2B).

### 2.3. Transplantation of hADC-mNPCs Promoted Endogenous mNPC Differentiation into Oligodendrocytes Following SCI

We examined neuronal (MAP-2), oligodendrocyte (Olig-2), and astrocyte (GFAP) marker protein expression at 1, 2, and 5 weeks after cell transplantation. Immunoblotting results found that MAP-2 protein expression was generally higher in the hADC-mNPC group than in the mNPC or pLXSN-mNPC groups (Figure 3A), and significantly so by 5 weeks post-transplantation (Figure 3B). At 2 weeks, GFAP expression was significantly higher in the mNPC and pLXSN-mNPC groups than in the hADC-mNPC group; while hADC-mNPC mice still expressed less GFAP at 1 and 5 weeks, this difference was not significant (Figure 3C). Moreover, at 2 weeks post-transplantation, GFAP expression decreased in hADC-mNPC mice compared with that in mNPC and pLXSN-mNPC mice (Figure 3C). Finally, Olig-2 expression increased over time in all groups. However, it was notably higher in the hADC-mNPC group than in the mNPC group at 1 week (Figure 3D).

### 2.4. Transplanted hADC-mNPC Colocalization with Oligodendrocytes Was Stronger Than pLXSN-mNPCs at Early Stage after SCI

Immunostaining in mNPC-, pLXSN-mNPC-, and hADC-mNPC-transplanted mice using the classical astroglia marker, glial fibrillary acidic protein (GFAP), and the oligodendrocyte marker, Olig-2, confirmed specific PKH-26-labeled expression in astrocytes and oligodendrocytes. 

The hADC-mNPC group had few GFAP+ with PKH-26 bivalent cells; however, the mNPC or pLXSN-mNPC group had GFAP+PKH-26 bivalent cells, which were observed at 2 weeks after SCI (Figure 4A).

At 2 weeks post-SCI, immunofluorescence staining for Olig-2 demonstrated that the PKH-26 expressing pLXSN-mNPC and hADC-mNPC group exhibited co-localization of Olig-2, whereas the mNPC group did not (Figure 4B). In the hADC-mNPCs group, an increase in the number of PKH-26+ with Olig-2 bivalent cells was observed compared to that in the pLXSN-mNPC group.

### 2.5. Transplantation of hADC-mNPCs Preserved and Enhanced Remyelination in the Injured Spinal Cord

In all experimental groups, SCI caused severe demyelination at the lesion site. Results from LFB staining indicated that hADC-mNPCs preserved myelination at 6 weeks after SCI, with that group possessing more myelin sheaths than the mNPC and pLXSN-mNPC groups (Figure 5A–C). Ultrastructural observations using TEM (×10,000) confirmed these findings. Additionally, TEM-based morphological examination showed that the hADC-mNPC group had significantly thicker myelin sheaths than the other two groups, which exhibited axonal profiles with poor myelination (Figure 5D–F). Thus, oligodendrocytes differentiated from hADC-mNPCs have the capacity to form mature myelin sheaths and remyelinate axons.

### 2.6. Transplantation of hADC-mNPCs Regulated BMP Expression in SCI Mice

We measured protein concentrations in tissue samples at 1, 2, and 5 weeks after transplantation to determine how BMP signaling changed (Figure 6A). Western blotting results showed that BMP-2 protein expression in the hADC-mNPC group increased significantly compared with levels in the mNPC group at 2 weeks post-transplantation (Figure 6B). Additionally, BMP-7 expression was higher in the hADC-mNPC group than in the pLXSN-mNPC and mNPC groups at 1, 2, and 5 weeks post-transplantation (Figure 6D). Conversely, the hADC-mNPC group had noticeably lower BMP-4 expression than both groups at all three time points (Figure 6C).

### 2.7. Transplantation of hADC-mNPCs Improved Locomotor and Bladder Functional Recovery Following SCI

At 1 day post-SCI, all animals exhibited complete hindlimb paralysis (BMS score = 0). At 1 week post-SCI, animals were either capable of slight ankle movements or none at all. 

At 1 week after transplantation, hADC-mNPC mice were able to step on the floor with the soles of their feet, whereas mNPC and pLXSN-mNPC mice rarely showed such behavior. By 3 weeks, the hADC-mNPC group showed occasional plantar stepping, while the mNPC and pLXSN-mNPC groups sometimes exhibited plantar placement without weight support, although walking remained impossible. The hADC-mNPC group also significantly differed from the control group in terms of extensive angle movement. 

At 4 weeks post-SCI, the hADC-mNPC group could not coordinate body movement but could occasionally rotate the top of the paw to take steps, whereas such movements were nearly absent in mNPC and pLXSN-mNPC mice. The hADC-mNPC group differed significantly from control and mNPC groups in this measure. 

At 5 weeks post-SCI, the hADC-mNPC group stepped parallel to the floor on initial contact, occasionally took steps coordinated with body movement, and rotated the back of the feet; in contrast, body movement was minimal in mNPC and pLXSN-mNPC mice, and walking continued to be absent. The differences between hADC-mNPC, pLXSN-mNPC, mNPC, and control groups were significant. 

At 6 weeks post-SCI, the hADC-mNPC group had recovered almost all control of body motion, stepping consistently by rotating the back of the foot, stepping parallel to the floor on initial contact, and exerting force at the tail end. The mNPC and pLXSN-mNPC groups occasionally performed plantar steps and moved their paws in coordination with the body. Overall, BMS scores and behavior differed significantly between the hADC-mNPC, pLXSN-mNPC, mNPC, and control groups (Figure 7A). 

The result of daily manual urine collection confirmed that all groups expelled a similar amount of urine from day 1 post-SCI to day 7. By 2 weeks, manually collected urine content was twice as low in the hADC-mNPC group as in the control, mNPC, and pLXSN-mNPC groups. These data demonstrate that bladder function returned more quickly to the hADC-mNPC group than to the other groups (Figure 7B). Urinary capacity returned to all groups after 2 weeks, and assisted urination was stopped. 

## 3. Discussion

Agmatine has well-studied neuroprotective properties [31,36,55,56,57] that could enhance the use of stem cell transplantation to treat CNS injuries, including SCI. Stem cell transplantation is intended to replace lost cells and provide trophic support to increase host neuron survival, along with host-mediated regeneration, repair, and plasticity [13,58,59]. Previously, we showed that agmatine enhanced neurogenesis of adult NPCs in the subventricular zone, and hADC transfection increased NPC differentiation [41,42,60]. Other studies have demonstrated that agmatine plays nerve protection and regeneration roles in various diseases [29,30,37,39,46,55,61,62,63]. Here, we followed up on our previous study to confirm the therapeutic effects of hADC-overexpressing NPCs on SCI.

We first selected an SCI phase for cell implantation. The three phases of SCI (based on elapsed time after injury and pathophysiological criteria) are acute, subacute, and chronic. The acute phase lasts about 48 h immediately after initial hemorrhage, physical damage, and vascular damage: the injury results in ion imbalance, neurotransmitter accumulation (excitation toxicity), inflammation, edema, bleeding, ischemia, and cell necrosis [64,65,66,67,68,69,70]. The subacute phase is characterized by a phagocytic response and reactive astrocytes; the latter causes stellate glial scar formation, which prevents nerve tissue regeneration and majorly impedes recovery [10,65,67,68,69,70,71,72]. Chronic SCI refers to the presence of symptoms for at least one year, along with a permanent cessation of neuronal impulse conduction in the spinal cord. Most such cases occur due to spinal cord deformation or vascular ischemia from trauma, tumors, and infections. During chronic SCI, nerve defects do not heal, leading to disorders such as convulsions, joint contractions, sensory inaction, and sphincter-movement abnormalities [67,69,72,73,74]. A characteristic of this phase is the development of a syrinx after scars have formed [65]. We chose the subacute stage for transplantation because nerves are more likely to regenerate in this earlier period than during the chronic stage. Furthermore, the subacute stage is when treatments such as cell transplantation are the most needed, and actual clinical trials largely consist of this patient subpopulation. Several studies have also shown that NSC transplantation in the subacute stage has the best therapeutic effect. We thus considered our experimental timing to be the most clinically effective and relevant. 

A week after transplantation, we confirmed that transplanted cells were concentrated in the lesion site. We also confirmed that the hADC-mNPC group had more transplanted cells at the damage site than the other two groups (mNPCs and pLXSN-mNPCs) (Figure 1A,B). In our previous study, we confirmed that agmatine expression under oxidative damage increased significantly in the hADC-hMSC group compared with reference agmatine expression [45]. Additionally, we have shown that transplanted hADC-hMSCs had higher survival, proliferation, and migration than hMSCs alone, leading to enhanced functional recovery after SCI [45]. Increased agmatine secretion in response to an inflammatory environment is thus the most plausible explanation for higher graft survival in the hADC-mNPC group after SCI.

Next, we evaluated whether hADC-mNPC transplantation promotes tissue repair in SCI. Glial scar formation decreased significantly in hADC-mNPC mice compared with that in mNPC and pLXSN-mNPC mice, indicating that hADC overexpression in mNPCs enhances tissue repair and explaining the functional improvement in these animals at 2 weeks after cell transplantation (Figure 2A,B). The hADC-mNPC group also expressed lower levels of reactive astrocyte proteins (Figure 3C). 

We examined the growth and differentiation of transplanted hADC-mNPCs to better understand how they promote functional recovery and tissue repair. We observed that neuronal markers markedly increased in the hADC-mNPC group at 5 weeks post-transplantation compared with the mNPC and pLXSN-mNPC groups (Figure 3B), a result confirmed through immunostaining (Figure 4C). Thus, hADC-mNPCs appeared to induce neuronal differentiation after they were transplanted. 

Differentiation to oligodendrocytes also occurred the most frequently among hADC-mNPCs (Figure 3D and Figure 4I), consistent with our previous study. After SCI, axon remyelination is important for functional recovery and is thought to depend on oligodendrocyte progenitor cells that give rise to nascent remyelinating oligodendrocytes [16,75,76,77]. Therefore, increasing the number of oligodendrocytes increases the likelihood of recovering locomotor function [78]. Our findings suggest that transplanted hADC-mNPCs induce the differentiation of oligodendrocytes rather than astrocytes. Multiple staining and imaging experiments revealed enhanced myelin sheaths in hADC-mNPCs at 6 weeks post-SCI, along with remyelinated or mature axons (Figure 5C,F). In contrast, the mNPC and pLXSN-mNPC groups possessed numerous degenerating myelinated axons and microglia/macrophage cells, along with noticeably fewer intact myelin sheaths (Figure 5D,E).

We confirmed changes to BMP-2/4/7 expression after cell transplantation following SCI. BMP-2/7 expression was higher in the hADC-mNPC group than in the other groups at 1, 2, and 5 weeks after cell transplantation. This difference was significant at 2 weeks post-transplantation, when BMP-4 expression was also significantly lower in hADC-mNPC mice. Our results corroborate previous research demonstrating that increased BMP-2/7 expression after agmatine administration significantly affects neuron and oligodendrocyte production [46]. Furthermore, BMP-4 inhibits astrocytes and increases oligodendrocytes, a function that can be mediated by hADC-generated agmatine [46,79]. We therefore propose that hADC affects BMP-4 regulation through agmatine production, reducing the formation of reactive astrocytes and glial scars after SCI and eventually playing a major role in axonal regeneration.

Functional recovery following SCI depends on myelin preservation and remyelination [80,81]. Here, we clearly observed that hADC-mNPC mice had more myelin sheaths at their lesion sites than the mNPC and pLXSN-mNPC mice. We therefore concluded that hADC-mNPCs promote the regeneration of disintegrating axons when implanted in the damaged spinal cord. We also used BMS scores to verify whether hADC-mNPC transplantation actually improves post-SCI functional recovery. Our results indicated that mouse BMS scores improved significantly at 3–4 weeks after hADC-mNPC transplantation compared with pLXSN-mNPC transplantation. By 5 weeks post-transplantation, hADC-mNPC mice were close to normal mice in terms of behavior. This improved locomotor function confirmed our prior results using agmatine treatment and hADC-hMSC transplantation after SCI [45,46,63].

Because SCI patients lose the ability to urinate, residual urine often remains to causes secondary infections and complications in the urinary tract and kidneys [78]. In SCI mice, self-voiding is similarly impossible up to 2 weeks after surgery. Here, we observed that the hADC-mNPC mice had less urine residue in the bladder and recovered self-voiding function faster.

Taken together, our behavioral findings are consistent with the hADC-hMSC results we previously reported, suggesting that hADC overexpression in mNPCs is a potential alternative for SCI treatment. To advance clinical applications, further research is needed to clarify the pharmacological mechanism of ADC in SCI. We also recommend that future studies provide a detailed analysis of upstream and downstream BMP pathways after hADC-hMSC transplantation, thus providing insight on the regulation of BMP signaling in human CNS disease.

## 4. Materials and Methods

### 4.1. Isolation and Cultivation of Murine-Cortex-Derived Neural Precursor Cells

Murine-cortex-derived NPCs (mNPCs) were cultivated following published methods [60]. The source of mNPCs was 14.5-day-old embryos (E14.5) (Koatech, Namyangju-si, Republic of Korea) extracted from placental tissue under a stereo microscope (SMZ-10, Nikon, Tokyo, Japan), using surgical kits (Fine Science Tools, Vancouver, BC, Canada). Cortices were aseptically dissected from brains and placed in Hank’s balanced salt solution (HBSS) (Gibco, Waltham, MA, USA). Tissues were triturated via repeated passage through a fire-polished constricted Pasteur pipette (Hilgenberg Pasteur pipette, Labdia, Malsfeld, Germany), allowed to settle for 3 min, and then transferred to a fresh tube and centrifuged at 1000× *g* for 5 min. The pellet was resuspended in NeuroCult basal medium (Stem Cell Technologies, Vancouver, BC, Canada) with NeuroCult proliferation supplement (Stem Cell Technologies, Vancouver, BC, Canada), 20 ng/mL recombinant epidermal growth factor (EGF) (Invitrogen, Carlsbad, CA, USA), and human basic fibroblast growth factor (bFGF) (Invitrogen, Carlsbad, CA, USA). Cells stained with trypan blue (Invitrogen, Carlsbad, CA, USA) were counted and then plated in a T-75 flask (BD Falcon, BD Biosciences, Franklin Lakes, NJ, USA) at a density of 2.5–3 × 10^6^ cells/mL. Cultures were maintained in a humidified atmosphere of 5 % CO_2_ at 37 °C. Culture media were replaced every 3 d. After 2 weeks, cultured mNPCs were used for subsequent experiments (Figure 8A).

### 4.2. Construction of Recombinant Retrovirus pLXSN Containing Human Arginine Decarboxylase Gene

Recombinant retroviral vectors (pLXSN) are multicloning sites that use a long terminal repeat (LTR) promoter to clone and regulate downstream genes. Following previous methods [41,42], retroviral pLXSN (K1060, Clontech, San Jose, CA, USA) vectors containing the recombinant hADC gene were transfected into mNPCs. First, full-length hADC cDNA (GenBank accession number AY325129) was PCR-amplified and ligated to pLXSN (Figure 8B). The vectors were then cloned in *E. coli* DH5α competent cells (Takara, Japan) and identified with restriction analysis. Next, hADC-expressing and empty pLXSN plasmids containing neomycin resistance genes were transfected into the retroviral packaging cell line PT67 (ATCC, UK) using Lipofectamine 2000 (Sigma-Aldrich, St. Louis, MO, USA) (Figure 8C). The optimal concentration for hADC and pLXSN resistant clone selection was achieved by adding G-418 (Sigma, USA) to Dulbecco’s modified Eagle’s medium (DMEM) (Gibco, Waltham, MA, USA) supplemented with 10% fetal bovine serum (FBS) (Gibco, USA). Cell cultures were maintained for 1 week in a 5% CO_2_ humidified atmosphere at 37 °C. To determine viral titers, virus-containing medium was first filtered through a 0.45 µm poly sulfonic filter (Sartorius AG, Bohemia, NY, USA), then added with a polybrene reagent (Sigma, USA) to the NIH/3T3 cell line (ATCC, UK) for infection. Clones with the highest titer were selected and stored at −70 °C until use [43]. After 1 week of culture, mouse-derived cortical NPCs (mNPCs) were infected with empty pLXSN and hADC-containing pLXSN. After 24 h of incubation with empty or hADC-containing pLXSN, the medium was replaced with the mNPC culture medium and maintained for another week. The subsequent experiments used mNPCs infected with hADC (hADC-mNPCs), mNPCs infected with pLXSN (pLXSN-mNPCs), and noninfected control mNPCs (Figure 8D). 

### 4.3. Animal Model of Compression SCI

Studies were conducted on male ICR mice (8 weeks old, 28 ± 5 g: Samtako, Osan, Republic of Korea). All animal experiments were performed in accordance with the Korean Food and Drug Administration guidelines. Protocols were approved by the Institutional Animal Care and Use Committee (IACUC) of Yonsei Laboratory Animal Research Center (YLARC) (permit number 2010-0350). All mice were maintained in the specific pathogen-free facility of the YLARC under controlled conditions (23 °C, 12 h:12 h light: dark cycle).

Mice were intramuscularly anesthetized with a combination of Zoletil 50 (0.6 mg/kg; Virbac, Carros, France) and Rumpun (0.4 mg/kg; Bayer, Leverkusen, Germany). Body temperatures were monitored with a rectal probe and maintained at 36.5–37.5 ℃ using heating pads. Thoracic vertebra (T) 8 to T10 was laminectomized without damaging the dura mater, and the spinal cord at T9 was subjected to compression (15 g/mm^2^) for 1 min using a bilateral microclamping clip (Fine Science Tools, Vancouver, BC, Canada). The bladder was manually pressed twice daily until spontaneous voiding occurred, and any hematuria or urinary tract infection was treated with ampicillin (1 mg/kg: Sigma, St. Louis, MO, USA) daily for 1 week. Food and water were freely accessible in the cages.

### 4.4. Cell Transplantation in SCI Mice

Model SCI mice were randomly divided into four groups: control (injury only, no transplantation; *n* = 20), mNPC transplantation (*n* = 30), pLXSN-mNPC transplantation (*n* = 30), and hADC-mNPC transplantation (*n* = 30) (Figure 8E). One week after SCI, cells (0.5 μL of 1 × 10^5^ cells/μL) were transplanted at T8 and T10. 

### 4.5. PKH-26 Labeling of Transplanted Cells

The mNPCs, pLXSN-mNPCs, and hADC-mNPCs were labeled with red fluorescent PKH-26 (2 × 10^−8^ mol/L culture medium; Sigma, St. Louis, MO, USA), following the manufacturer’s protocol. Briefly, detached cells were washed with serum-free medium and resuspended in 1 mL of dilution buffer. The cell suspension was mixed with an equal volume of labeling solution containing PKH-26 and incubated for 5 min at 23 °C. The fluorescent dye has an aliphatic reporter molecule that integrates into the cell membrane via selective partitioning (Figure 9A). After 2 mL of serum was added to terminate the reaction, cells were washed three times with culture medium and observed under a fluorescent microscope (CKX53, Olympus, Westborough, MA, USA).

### 4.6. Locomotor Recovery and Bladder Function Assessments

Hindlimb locomotor recovery was assessed in an open field test using the nine-point Basso Mouse Scale (BMS) [82]. Scores 0–2 reflect complete absence of ankle movement to greater ankle movement; scores 3–4 correspond to improvement in step and plantar placement; scores 5–8 corresponds to paws in standing position, hindlimb–forelimb coordination, and trunk stability; and score 9 indicates normal locomotion with trunk stability. The final BMS score was the average of each group member’s left and right hind legs. For 1 week before surgery, all animals were exercised once a day at the same time to confirm normal mobility. Additionally, BMS tests were performed weekly by two scorers blinded to experimental conditions, starting from 1 week before SCI and ending at 6 weeks after SCI. Starting from SCI until 14 d postinjury (DPI), bladders were manually stimulated twice daily (*n* = 10 per group) until mice regained normal autonomic bladder function (approximately 11–15 DPI). Retained urine from each mouse was collected and measured, both in the morning and evening sessions (12 h interval) until 14 DPI (Figure 9B).

### 4.7. Glial Scar Formation Analysis

To quantify glial scar area in the mNPC, pLXSN-mNPC, and hADC-mNPC mice, tissue sections (20 mm) were obtained 2 weeks after SCI and sequentially immunoreacted with GFAP antibody (1:500, Thermo, Waltham, MA, USA) at 4 °C overnight. Subsequently, tissue sections were incubated with the appropriate biotinylated secondary antibodies. Immunostaining was performed using an ABC kit (Vector, Burlingame, CA, USA), followed by reaction with 3,39-diaminobenzidine tetra hydrochloride (DAB, Sigma, St. Louis, MO, USA). Negative controls lacked the primary antibody. Glial scar area was obtained from measuring GFAP-positive regions around the lesion site (with central cavity and the number of reactive astrocytes) in Image J (National Institutes of Health, Bethesda, MD, USA).

### 4.8. Immunohistochemistry and Histology

Mice were deeply anesthetized before saline perfusion. Extracted tissue (spinal cord sections T8, T9, and T10) was subjected to histological analysis at 2, 3, and 6 weeks after SCI. Dissected spinal cords were postfixed overnight in 4% paraformaldehyde (PFA), incubated in 30% sucrose prepared with phosphate-buffered saline (PBS) solution at 4°C for 3 d, embedded in Tissue-Tek OCT compound (Japan), and sectioned longitudinally using an UV cryostat (Leica Microsystems, Wetzlar, Germany). The frozen sections (18–20 μm, *n* = 5 per group) were air-dried at RT for 20 min and incubated with blocking solution at 37 °C for 1 h. Next, sections were incubated overnight at 4 °C with the following primary antibodies: mouse anti-MAP-2 (1:500, Sigma-Aldrich, Burlington, MA, USA), mouse anti-GFAP (1:500; Thermo, Waltham, MA, USA), and goat anti-Olig-2 (1:250; Santa Cruz Biotechnology, USA). Slides were then washed three times in PBS and incubated at 37 °C with either FITC- (1:500; Chemicon, Nuremberg, Germany) or rhodamine- (1:500; Chemicon, Nuremberg, Germany) conjugated secondary antibodies for 2 h. Finally, slides were stained with DAPI (Vector, Olean, NY, USA) to visualize nuclei under a confocal microscope (LSM 700; Carl Zeiss, Oberkochen, Germany).

### 4.9. Immunoblotting

Spinal cord tissues (T8, T9, and T10) were dissected to remove meninges and nerve roots at 1, 2, and 5 weeks after cell transplantation. Tissues were homogenized on ice in 500 μL of radio immunoprecipitation assay (RIPA) buffer containing 150 mM sodium chloride, 1% Nonidet-P40 (NP-40), 0.5% sodium deoxycholate, 0.1% sodium dodecyl sulfate (SDS), 50 mM Tris (pH 8.0), phosphatase inhibitor solution, 20 mM Tris-HCl (pH 7.5), 2 μg/mL aprotinin, 5 μg/mL leupeptin, 1 μg/mL pepstatin-A, 1 mM phenylmethylsulphonyl fluoride (PMSF), 5 mM ethylenediaminetetraacetic acid (EDTA), 1 mM EGTA, 5 mM sodium fluoride (NaF), and 1 mM sodium orthovanadate (Na_3_VO_4_). Equal amounts of protein (50 µg) were loaded onto 8–10% polyacrylamide gels and transferred onto polyvinylidene fluoride (PVDF) membranes. Membranes were then blocked with 5% skim milk in Tris-buffered saline with 0.1% Tween 20 detergent (TBS-T) for 1 h at 37 °C, then incubated overnight at 4 °C with the following primary antibodies: mouse anti-MAP-2 (1:1000; Sigma-Aldrich, USA), mouse anti- GFAP (1:1000; Thermo, USA), goat anti-Olig-2 (1:1000; Santa Cruz, Dallas, TX, USA), mouse anti-BMP-2 (1:1000; Abcam, Cambridge, UK), mouse anti-BMP-4 (1:250; Santa Cruz, USA), and rabbit BMP-7 [1:1000; Abcam, UK]. Membranes were washed with TBS-T before being incubated with horseradish peroxidase-conjugated secondary antibody (1: 2000; Chemicon, Nuremberg, Germany) at 37 °C for 2 h. Subsequently, membranes were incubated in ECL (Super Signal West Pico PLUS Chemiluminescent substrate) (Thermo, USA) following manufacturer protocol. Immunoreactive bands were digitally scanned and analyzed in ImageJ (National Institutes of Health, USA). To control for protein loading, membranes were probed with mouse anti-β-actin (1:3000, Abcam, UK), and protein concentrations were normalized to β-actin.

### 4.10. Luxol Fast Blue Staining

Myelinated tissue was visualized using Luxol fast blue (LFB) staining. Spinal cord tissue sections (*n* = 5 per group) were rinsed in PBS and serially dehydrated in ethanol solutions (70%, 95%, and 100%) for 30 min each. Sections were then placed in 0.1% LFB solution (Sigma, USA) for oven incubation at 56 °C for 4 h. Excess stain was rinsed with 95% ethanol. To differentiate staining, tissue sections were incubated in 0.05% lithium carbonate solution and counterstained with 0.1% cresyl violet solution (Sigma, USA) for 30 s. 

### 4.11. Assessing Ultrastructural Spinal Cord Changes

Post-SCI microstructural changes in the myelin sheath were assessed with transmission electron microscopy (TEM). Briefly, mice were perfused with normal saline, followed by a solution containing 2% glutaraldehyde and 4% PFA. After thermal stress for 12 h, each sample was fixed with 2% glutaraldehyde-PFA in 0.1 M PBS for 2 h and washed three times for 30 min in 0.1 M PBS. They were then postfixed with 1% OsO_4_ dissolved in 0.1 M PBS for 2 h, dehydrated in an ascending ethanol series (50–100%), and infiltrated with propylene oxide. Specimens were embedded using a Poly/Bed 812 Embedding Kit (Polysciences, Warrington, PA, USA) and polymerized at 60 °C in an electron microscope oven (TD-700, DOSAKA, Japan) for 24 h. After incubation, sections (350 nm) were sliced and stained with toluidine blue to confirm embedding quality under a light microscope. Thinner [70 nm] sections were then sliced in a LEICA Ultracut UCT Ultra-microtome (Leica Microsystems, Germany) and counter-stained with 7% uranyl acetate and lead citrate for 20 min. These sections were observed under a TEM (JEM-1011, JEOL, Tokyo, Japan) at an acceleration voltage of 80 kV.

### 4.12. Statistical Analysis

All data are presented as means ± standard error. One-way analysis of variance (ANOVA) with post hoc Tukey’s HSD test was used to determine between-group variation in SPSS 18.0. Significance was set at *p* < 0.05.

## 5. Conclusions

Our results suggest that hADC gene overexpression is an effective way to enhance the therapeutic potential of cell therapy for SCI. Transplanting hADC-mNPCs in an SCI mouse model improved locomotor and bladder function, decreased initial glial scar formation, induced remyelination via oligodendrogenesis, and increased neuronal differentiation of transplanted cells. To improve the degree and speed of functional recovery, the optimal stage for cell transplantation must be determined through further research. In particular, evidence from comparative data will be extremely useful. Future studies should also confirm the effects of ADC overexpression through different means, such as transplanting other virus types or even nonviruses that include the ADC gene.

## Figures and Tables

**Figure 1 ijms-23-15784-f001:**
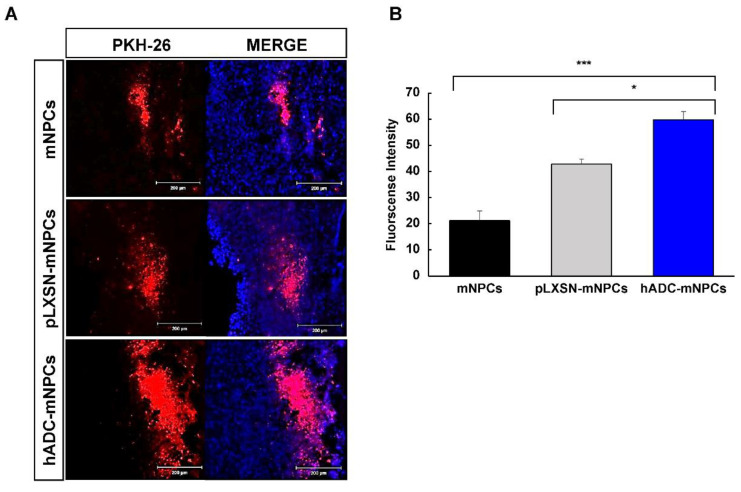
Migration and survival of mNPCs around the lesion site at 1 week after cell transplantation following SCI. (**A**) A higher number of PKH-26-labeled hADC-mNPCs grafted found around the lesion area (T9) than there were mNPCs and pLXSN-mNPCs at 1 week after cell transplantation. (**B**) More PKH-26-labeled hADC-mNPCs (60% fluorescence intensity) were present around the lesion area (T9) than there were mNPCs (20%) and pLXSN-mNPCs (40%). Data are presented as means ± standard error of mean (SEM). (*n* = 5 per sample, * *p* < 0.05, *** *p* < 0.001).

**Figure 2 ijms-23-15784-f002:**
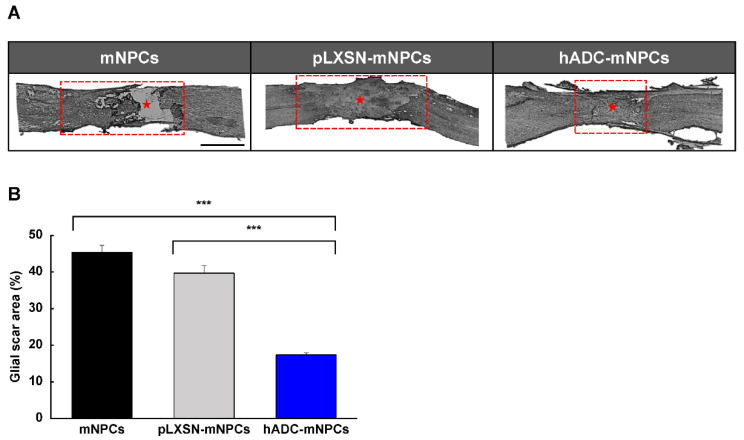
hADC overexpression in mNPCs reduces glial scar volume 2 weeks after transplantation into compression-lesioned spinal cords of adult mice. (**A**) Immunohistochemistry of longitudinal spinal cord sections stained with antibodies against GFAP at T8, T9, and T10 at 2 weeks after transplantation of hADC overexpressing mNPCs (hADC-mNPCs), empty retroviral overexpression mNPCs (pLXSN-mNPCs), or mNPCs alone. The red box was the site of the glial scar lesion, and the middle of the glial scar is indicated by a red asterisk. Scale bar = 5 mm. (**B**) Lesion areas are significantly larger in mice with mNPC transplantation (45% of the area shown in part A) than transplantations of pLXSN-mNPCs (40% of the area shown in A) and hADC-mNPCs (19% of the area shown in A). Data are presented as means ± SEM. (*n* = 5 per sample, *** *p* < 0.001).

**Figure 3 ijms-23-15784-f003:**
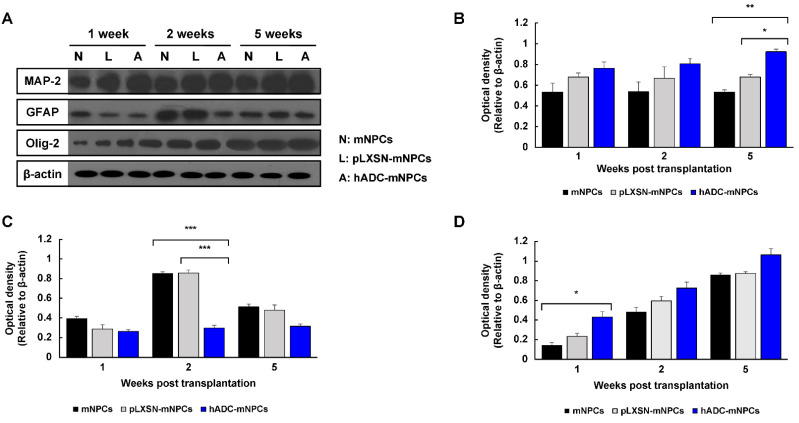
Western blotting of MAP-2, GFAP, Olig-2, and β-actin from T8, T9 (lesion area), and T10 at 1, 2, and 5 weeks after cell transplantation. (**A**) Representative Western blots of MAP-2, GFAP, Olg-2, and β-actin. (**B**) Quantification of MAP-2, showing that hADC-mNPC mice had significantly higher expression than pLXSN-mNPC or mNPC mice at 5 weeks post-transplantation. (**C**) Quantification of GFAP, showing a significant decrease in the hADC-mNPC group compared with the pLXSN-mNPC and mNPC groups at 2 weeks post-transplantation. (**D**) Quantification of Olig-2, showing a significant increase in the hADC-mNPC group compared with the pLXSN-mNPC and mNPC groups at 1 week post-transplantation. Data are presented as the mean ± SEM. (*n* = 5 per sample, * *p* < 0.05, ** *p* < 0.01, *** *p* < 0.001).

**Figure 4 ijms-23-15784-f004:**
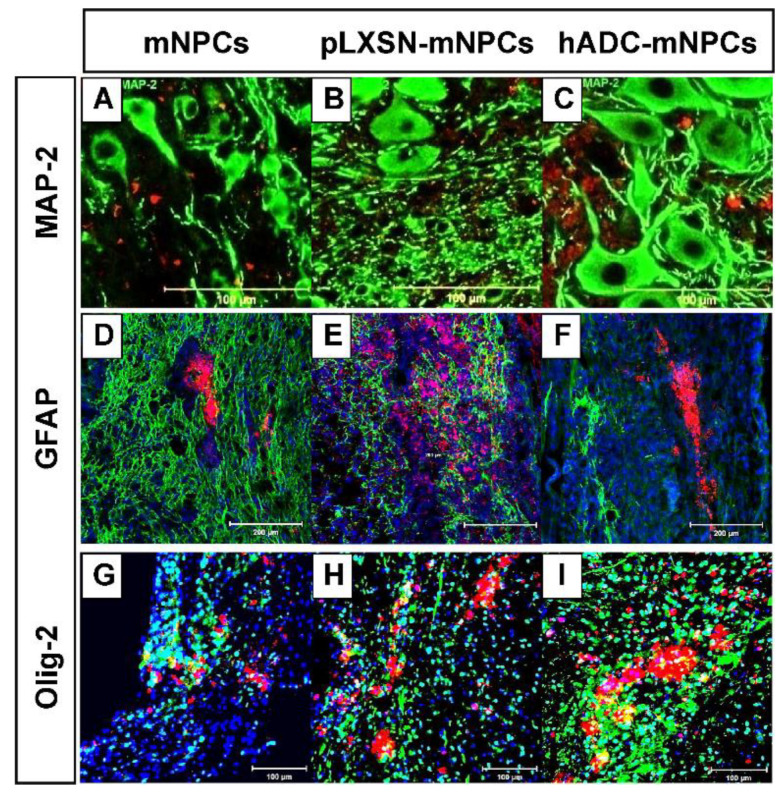
Double immunohistochemistry of PKH-26-labeled mNPCs, pLXSN-mNPCs, and hADC-mNPCs, along with the marker microtubule associated protein-2 (MAP-2), fibrillary acidic protein (GFAP), and Olig-2 around lesion sites after SCI. (**A**–**C**) Immunofluorescence staining of MaP-2 (green) in PKH-26-labeled mNPCs, pLXSN-mNPCs and hADC-mNPCs (red) at 5 weeks post-transplantation. (**D**–**F**) Immunofluorescence staining of GFAP (green) in PKH-26-labeled mNPCs, pLXSN-mNPCs and hADC-mNPCs (red) at 1 week post-transplantation. DAPI staining (blue) indicating the nuclei. (**G**–**I**) Immunofluorescence staining of Olig-2 (green) in PKH-26-labeled mNPCs, pLXSN-mNPCs, and hADC-mNPCs (red) at 1 week post-transplantation. DAPI staining (blue) indicating the nuclei. Scale bar = 100 μm (MAP-2 and Olig-2), 200 μm (GFAP).

**Figure 5 ijms-23-15784-f005:**
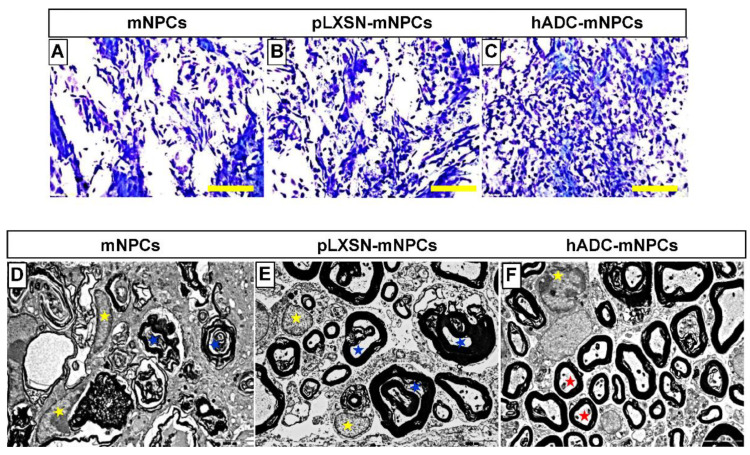
Luxol fast blue staining of T9 spinal cord sections obtained from the mNPC (**A**), pLXSN-mNPC (**B**), and hADC-mNPC (**C**) transplantation groups at 6 weeks after SCI. The hADC-mNPC group (**C**) had smaller cystic cavities and more myelin sheaths than the other two groups (**A**,**B**). Scale bar = 20 μm. Transmission electron microscopy (TEM, 10,000×) of the lumbar section where mNPCs (**D**), pLXSN-mNPCs (**E**), and hADC-mNPCs (**F**) were transplanted after SCI. Transplanted hADC-mNPCs promoted axon remyelination at the lesion site. Red stars indicate remyelinated or mature axons in the hADC-mNPC group, while blue and yellow stars indicate degenerating myelinated axons and microglia/macrophage cells, respectively.

**Figure 6 ijms-23-15784-f006:**
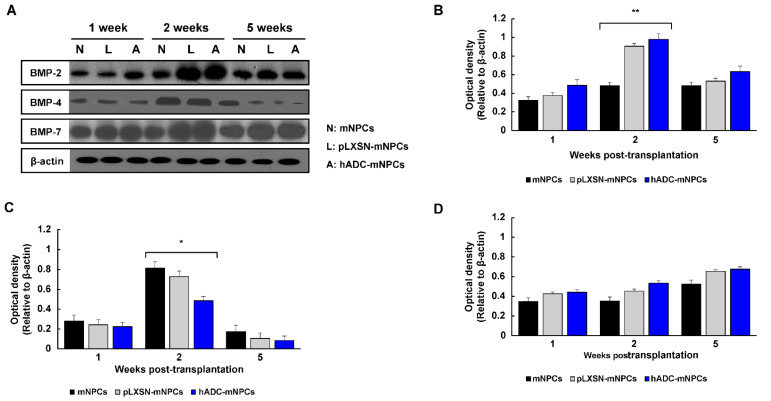
Western blot analysis of BMP-2, BMP-4, BMP-7, and β-actin from T8, T9, and T10 at 1, 2, and 5 weeks after cell transplantation. (**A**) Representative Western blots of BMP-2, BMP-4, BMP-7, and β-actin. (**B**) Quantification of BMP-2 showing a significant increase in hADC-mNPC mice compared with pLXSN-mNPC and mNPC mice at 2 weeks post-transplantation. (**C**) Quantification of BMP-4 showing a significant decrease in hADC-mNPC mice compared with pLXSN-mNPC and mNPC mice at 2 weeks post-transplantation. (**D**) Quantification of BMP-7 demonstrating a lack of significant difference across the three groups, despite slightly higher expression in the hADC-mNPC group at 1, 2 and 5 weeks post-transplantation. Data are presented as means ± SEM. (*n* = 5 per sample, * *p* < 0.05, ** *p* < 0.01).

**Figure 7 ijms-23-15784-f007:**
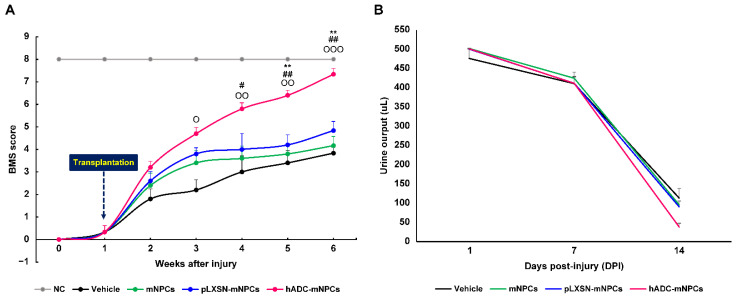
Mice with hADC-overexpressing mNPCs (hADC-mNPCs) exhibit improved motor performances and bladder function recovery. (**A**) Basso Mouse Scale (BMS) score up to 6 weeks after SCI, demonstrating continuous improvement in the hADC-mNPC group compared with other groups at 4, 5, and 6 weeks. (**B**) Comparison of bladder residual urine volumes up to 14 days post-SCI, showing that levels were lower in hADC-mNPC mice than in other mice. Data are presented as means ± SEM. (*n* = 10−40 per group, ^O^
*p* < 0.05, ^OO^
*p* < 0.01, ^OOO^
*p* < 0.001 (hADC-mNPCs versus vehicle), ^#^
*p* < 0.05, ^##^
*p* < 0.01 (hADC-mNPCs versus mNPCs), ** *p* < 0.01 (hADC-mNPCs versus pLXSN-mNPCs).

**Figure 8 ijms-23-15784-f008:**
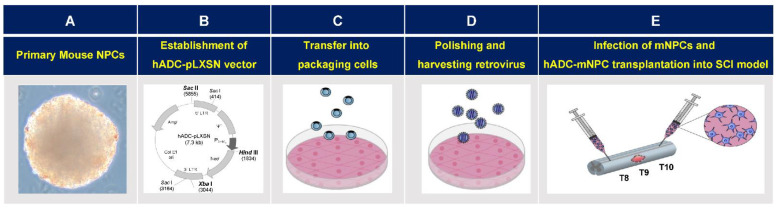
Retroviral vector and hADC-mNPCs therapy bioprocessing for spinal cord injury (SCI). (**A**) Isolation of primary mouse neural precursor cells (mNPCs) from brains of ICR mouse embryos, followed by neurosphere cultivation. (**B**) Full-length hADC cDNA was PCR-amplified and ligated to recombinant retroviral expression vector pLXSN. (**C**) The hADC-expressing pLXSN plasmids and empty pLXSN plasmids containing neomycin resistance genes were transfected using Lipofectamine 2000 into the retroviral packaging cell line PT-67. (**D**) Supernatants containing hADC genes and empty retrovirus pLXSN were infected to NIH-3T3 cells using polybrene reagent to determine viral titers. Retrovirus containing human arginine decarboxylase genes (hADC) were transfected into mouse neural precursor cells (mNPCs). (**E**) Spinal cords were compressed at thoracic vertebra (T) 9 (indicated by irregular pink shape representing glial scar). The experiments used mNPCs infected with hADC (hADC-mNPCs), mNPCs infected with pLXSN (pLXSN-mNPCs), and non-infected control mNPCs. Mice with SCI were randomly divided into four experimental groups and cells were transplanted into two regions—rostral (Thoracic 8) and caudal (Thoracic 10)—from the lesion site (T9) at 1 week after SCI.

**Figure 9 ijms-23-15784-f009:**
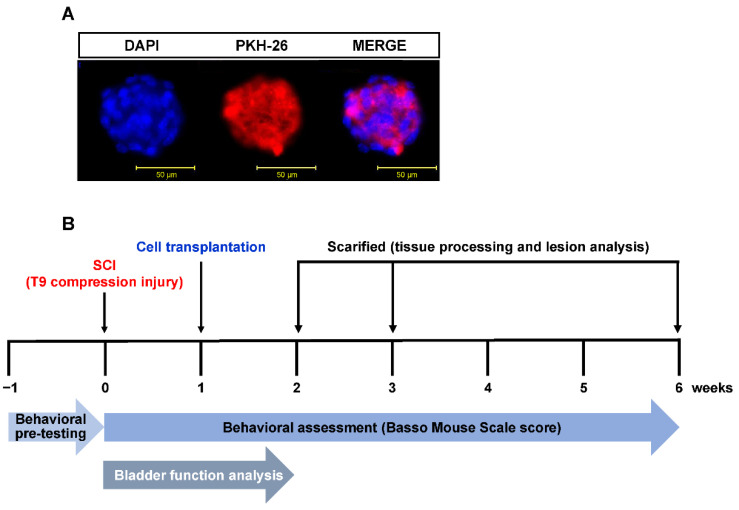
Experimental time schedule for in vivo SCI model of cell transplantation, behavioral assessment, bladder functional analysis, and tissue harvest. (**A**) mNPC labeling with the fluorescent dye PKH-26 (red); cellular nuclei were stained with DAPI (blue). (**B**) All mNPC, pLXSN-mNPC, and hADC-mNPC transplantations were performed 1 week after SCI. Transplanted cell retention was assessed in spinal cord explants at 1, 2, and 5 weeks after cell transplantation. Assessments of forearm function were performed before and after injury, and weekly following transplantation. Bladder function analysis was performed simultaneously twice a day until 2 weeks after SCI.

## Data Availability

The data used to support our findings are included within the article.

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
