# Peer review of "Neural Stem Cells Overexpressing Arginine Decarboxylase Improve Functional Recovery from Spinal Cord Injury in a Mouse Model"

_ijms, 2022, doi:10.3390/ijms232415784_

Round 1

Reviewer 1 Report

Authors investigated the effects of hADC-mNPCs against spinal cord injury based on behavioral, morphological, and biochemical studies. The results are interesting, but there are few concerns to consider for publication.

In figure 2, authors should demonstrate the high magnified microphotographs to show the morphology of GFAP-immunnoreactive scar. In addition, I suggest that authors should show migration and survival of PKH-26-labeled hADC-mNPCs demonstrated in Figure 1 with GFAP immunoreactive scars.

Although authors showed remyelination of cross-sectioned tissues, authors should demonstrate immunohistochemical data of myelin basic protein or proteolipid protein to exclude the OPC or pre-OL.  

Author Response

Comments and Suggestions for Authors

  1. Authors investigated the effects of hADC-mNPCs against spinal cord injury based on behavioral, morphological, and biochemical studies. The results are interesting, but there are few concerns to consider for publication.

  1. In figure 2, authors should demonstrate the high magnified microphotographs to show the morphology of GFAP-immunoreactive scar.

Response: I have attached a high magnification image.

In addition, I suggest that authors should show migration and survival of PKH-26-labeled hADC-mNPCs demonstrated in Figure 1 with GFAP immunoreactive scars.

Response: I have added a picture of the cell migration path that I transplanted.

  1. Although authors showed remyelination of cross-sectioned tissues, authors should demonstrate immunohistochemical data of myelin basic protein or proteolipid protein to exclude the OPC or pre-OL.

Response: Myelin basic protein (MBP) is the major structural element of CNS myelin, in which it is essential for both myelin wrapping and compaction (Readhead et al., 1987).

Unfortunately, I was unable to find the tissue slide because this experiment was already completed a long time ago. However, I found the result of the IF instead and have attached it.

(In figure) At 6 weeks after SCI, the mNPC and pLXSN-mNPC groups showed denuded axons as well as degraded myelin. In the hADC-mNPC group, a proportion of axons (examples denoted with yellow arrowheads) was associated with the reappearance of myelin rings.

Reviewer 2 Report

This study aimed to observe the therapeutic effect of hADC gene over-expression in murine brain-derived neural progenitor cells (mNPC) on spinal cord injury and its underlying mechanisms. This study is interesting, but the authors need to address the following concerns before the paper is ready for publication.

Major

1. Please provide clear GFAP staining images for Figure 2A.

2. The image scales in Figure 4 are not consistent, so the experimental results cannot be supported.

3. The description of BMP7 expression in the results is inconsistent with the figure legend and discussion.

4. The authors mentioned in the manuscript that "A week after transplantation, we observed that transplanted cells were concentrated in the lesion site but were largely absent along the site's boundaries and in the intact area [76]", however, the reference [76] cited here is not the findings of the authors.

5. Statistical analysis of some experimental results should be analyzed using two-way rather than one-way ANOVA.

Minor

1. Some references are missing. For example, "Other studies have demonstrated…... [64]".

Author Response

Comments and Suggestions for Authors

This study aimed to observe the therapeutic effect of hADC gene over-expression in murine brain-derived neural progenitor cells (mNPC) on spinal cord injury and its underlying mechanisms. This study is interesting, but the authors need to address the following concerns before the paper is ready for publication.

Major

  1. Please provide clear GFAP staining images for Figure 2A.

Response: I have attached a high magnification image.

  1. The image scales in Figure 4 are not consistent, so the experimental results cannot be supported.

Response: I have modified the corresponding figure.

  1. The description of BMP7 expression in the results is inconsistent with the figure legend and discussion.

Response: I agree with your advice and have changed the text and figure legend to highlight the result of BMP7.

  1. The authors mentioned in the manuscript that "A week after transplantation, we observed that transplanted cells were concentrated in the lesion site but were largely absent along the site's boundaries and in the intact area [76]", however, the reference [76] cited here is not the findings of the authors.

Response: When we rearranged the references according to the body order, the number was disrupted. I have checked all the references and changed them.

  1. Statistical analysis of some experimental results should be analyzed using two-way rather than one-way ANOVA.

Response: Thank you for your advice; I have changed the text accordingly.

The results measured in this experiment included the mean and standard deviation for each item, which were calculated using the SPSS version 18. program. One-way ANOVA was used for protein expression and intergroup differences, and two-way repeated measures ANOVA was used for behavioral measurement analysis.

Minor

  1. Some references are missing. For example, "Other studies have demonstrated…... [64]".

Response: When we rearranged the references according to the body order, the number was disrupted. I have checked all the references and changed them.

Reviewer 3 Report

This is a very interesting, well done, and well written paper. The authors have studied in a very detail the effect of Arginine decarboxylase exogenously  expressed in the neuronal cells. The work contains all proper controls. Some methods are difficult to fulfill. The results are confident and give a hope to use this approach in clinics. 

There are no drawbacks

Author Response

Comments and Suggestions for Authors

This is a very interesting, well done, and well written paper. The authors have studied in a very detail the effect of Arginine decarboxylase exogenously expressed in the neuronal cells. The work contains all proper controls. Some methods are difficult to fulfill. The results are confident and give a hope to use this approach in clinics. There are no drawbacks.

Response: Thank you for reviewing and acknowledging our paper.

Round 2

Reviewer 2 Report

Although the authors answered most of the questions in a satisfactory way, the authors did not fully address the second question. Please adjust the images in Figure 4 to the same scale. For example, the image scale of MAP-2 is different from that of Olig-2, and the magnification ratio of the former is significantly larger than that of the latter.